e-science/complexity

economic network, corporate reputation, stakeholder behaviour

**Author for correspondence:**
Frank Schweitzer
e-mail: fschweitzer@ethz.ch

# The interdependence of corporate reputation and ownership: a network approach to quantify reputation

## Yan Zhang and Frank Schweitzer

Chair of Systems Design, ETH Zurich, Weinbergstrasse 58, 8092 Zurich, Switzerland

(iD) FS, 0000-0003-1551-6491

We propose a novel way to measure the reputation of firms by using information about their ownership structure. Supported by the signalling theory, we argue that ownership relations channel reputation spillovers between shareholders and their invested companies. We model such reputation spillovers by means of a simple dynamics that runs on the ownership network, constructed from available databases. We focus on the core of the global ownership network with 1300 firms and 12 100 ownership links. Our method assigns an ownership-based reputation value to each firm, used to provide a quantitative reputation ranking. We compare our ranking with alternative rankings, to confirm that the top-ranked firms are correctly identified. We also demonstrate that our reputation measure does not correlate substantially with operating revenue or control and thus provides additional information about firms.

## 1. Introduction

The topic of corporate reputation has attracted considerable attention from both academics and practitioners over the last years. On the *academic* side, various conceptual models to quantify corporate reputation have been proposed in the field of marketing and management [1–4]. One representative example is the *reputation quotient*, a multidimensional construct that aggregates various dimensions of stakeholders' perceptions based on survey data [5]. On the *practitioners'* side, rating agencies have contributed *periodic ratings* of reputation, and many large companies have adopted practices for reputation management.

Our careful review of the literature and the industry practices shows that the current research on quantifying and measuring corporate reputation primarily uses survey data as the basis for reputation [3–5]. To extend this empirical basis, we take a different approach by considering data on ownership relations that connect firms. Our enlarged perspective is motivated by the following two arguments: First, the reputation of a company does not only depend on itself but also on the actions and reputations of other companies. This is illustrated by the phenomenon of *reputation spillover*, where reputation changes in one company affect the reputation of others through inter-firm relations, such as ownership. Second, corporate shareholders, one of the most important stakeholder groups of a company, are so far largely neglected when quantifying reputation. We know of only a few works that focus on corporate shareholders and the corresponding ownership structure [6] with respect to reputation. This is at difference to other types of stakeholders that have been adequately addressed, such as the public [7] and opinion leaders [8].

Therefore, in this paper, we explore how the reputation of a company depends on its corporate shareholders, specifically, and its ownership structure, in general. To this end, we adopt the signalling theory from social science [9,10], to relate ownership structures and reputation spillovers. In order to model the complex ownership relations among firms, we use methods from network analysis. As an empirical basis, we use the ORBIS database to construct the global ownership network, in which nodes represent firms and links ownership relations. Furthermore, we propose a dynamics for the reputation of firms conditional on the reputation of their owners. Because the interaction topology underlying the dynamics is given by the ownership network, we are dealing with a network of two layers: one layer describes ownership relations, the other reputation relations. Our work uses this interdependence between corporate reputation dynamics and ownership network, to provide a quantitative method to proxy reputation.

# 2. Theoretical background

## 2.1. Notion of reputation and ownership

Several definitions of corporate reputation have been formulated in the literature. Fombrun & Shanley [11] interpreted reputation as 'the outcome of a competitive process in which firms signal their key characteristics to constituents to maximize their social status.' Wartick [12] extended the concept and defined it as 'aggregation of a single stakeholder's perception of how well organizational responses are meeting the demands and expectations of many organizational stakeholders'. Similarly, Brammer & Millington [13] provided a simplified definition of reputation as 'the collective opinion of an organization held by its stakeholders'. Despite variations in the wording, the above definitions agree in that corporate reputation is based on stakeholders' *perceptions* about a firm's status and ability over time.

Because of the different identities of stakeholders, reputation is a multidimensional construct [4,14] and cannot be directly measured. In our work, we focus on the dimension of reputation related to *corporate shareholders*, a key stakeholder group of companies, which are connected by *ownership relations*. These define the ownership structure of a firm, which characterizes the distribution of cash-flow rights and decision rights among its shareholders, who can either be individuals or institutions. Ownership relations can be further used by companies 'to hide assets, to restructure assets, to gain access to knowledge, to increase legitimization, and to secure supplies and markets' [15]. As one of the few sources that channel informational cues and generate expectations about future performance, ownership structure, particularly institutional ownership, has been recognized as a determinant influencing corporate reputation [16].

In this direction, one line of research built and tested theories that describe the correlation between ownership structure and corporate reputation. Particularly, using a dataset of 292 large US firms, Fombrun & Shanley [11] combined economic and sociological approaches to study firms' interactions with the public. They related assessments of reputation to informational signals. By means of cross-sectional time-series analysis, they demonstrated a positive effect of the concentration of ownership in institutional investors on reputation. In the same direction, Brammer & Millington [13] distinguished between signals concerning companies that the public may receive, and identified the determinants of corporate reputation with a sample of large UK firms. They reported a similar positive correlation between corporate reputation and long-term institutional shareholders.

The above works mainly concern firms in common-law countries, where firms have a dispersed ownership structure and the manager–shareholder relation is the main source of conflict. In

comparison, using archived data on the top 100 reputed firms in the civil-law country Spain, Delgado-Garcia *et al.* [6] examined the impact of ownership structure on corporate reputation. They focused on several features of ownership structure that are most visible to stakeholders, and reported a negative correlation between the concentration of ownership in the largest shareholder and corporate reputation. This negative correlation was further explained by the fact that in civil-law countries, large shareholders are more common and can use their voting power to extract private benefits. All these works confirm that the legal and institutional setting of a country influences ownership structure, which further impacts the correlation between ownership and reputation.

Another line of research explored how *reputation spillovers* happen in the context of ownership. Reputation spillover, or reputation contagion, describes the dynamic process in which reputation changes in one company lead to the gain or loss of reputation in other companies. The underlying logic for this line of research is the following: Ownership not only grants shareholders potential profits and voting power but also creates feedback loops determining the extent to which shareholders bear additional costs and other benefits. Concretely, shareholders can monitor and exert control over the managerial performance of a company to align their interests.

As a result, *reputation damage* to one company may be interpreted as inefficient monitoring exercised by shareholders, which further leads to negative spillover effect in the reputation of the shareholding companies. For example, Saapar & Soussa [17] discussed the risk of internal reputation contagion in the conglomerization process, i.e. mergers and acquisitions, across financial services industries. Considering three organizational structures that combine banking and non-banking activities differently, they argued that there is a trade-off between efficiency in cost savings and the likelihood of reputation contagion. Also, Kang [18] studied the effect of director interlocking, i.e. a person on the corporate board of one firm is also affiliated with another firm, on the spillover effect of reputational loss. They reported that the likelihood of negative spillover is reduced by institutional shareholders like mutual funds and public pension funds, as a result of independent and active monitoring.

One limitation of the above studies [6,11,13,17,18] is that they look only at the *direct* shareholding relations with *sample data* of *selected* companies at the *country* level. This will be improved in our paper, considering also indirect shareholding relations from a large-scale dataset of 1318 companies in 26 countries. Nevertheless, in spite of the relatively small scales of their analysis, the mentioned studies provide the insights that the identity of shareholders influences corporate reputation, and reputation spillovers can be channelled through ownership relations.

We build on such insights by developing a comprehensive network perspective as a new way to understand the interdependency between reputation and ownership relations. Here, we emphasize that we do not aim to provide a new notion of reputation. Instead, we focus on the dimension of reputation that relates to corporate shareholders, a key stakeholder group of companies connected by ownership relations.

The network perspective has been extensively applied to inter-organizational contexts [19], such as R&D alliance [20], credit [21] and director interlocking [22]. The literature can be roughly categorized into two directions. One direction is to link properties of empirically observed networks to organizational outcome. For example, in a longitudinal study of 1106 firm, Schilling & Phelps [20] explored how the structure of alliance networks influences the potential for knowledge creation in inter-firm networks. They identified key network properties that can significantly impact the innovative output of firms in the network. Similarly, using the ORBIS dataset that describes ownership relations of firms, Vitali *et al.* [23] looked into the structure of the global ownership network and confirmed that a large portion of control belongs to a small set of financial institutions. The other direction is to explain organizational behaviour based on network interaction and to classify possible outcomes. For example, König *et al.* [24] considered a network formation process in which firms endogenously choose their partners to create innovation. They identified under which conditions this process can lead to stable network structures that match the properties of empirical R&D networks.

## 2.2. Firm relations as a multilayer network

To represent the interdependency between reputation relations and ownership relations, we have chosen the framework of *multilayer networks*. A network, or a graph in mathematical terms, consists of *nodes* that are connected by *links*. The nodes in our case represent the *firms*. As firms can have different types of relations, we depict each type of relation in a separate network *layer* (figure 1). In a multilayer network, the nodes in each layer are identical, but the links representing their relations are usually different. In our case, we have a network of two layers: *layer 0* contains the *ownership relations* between

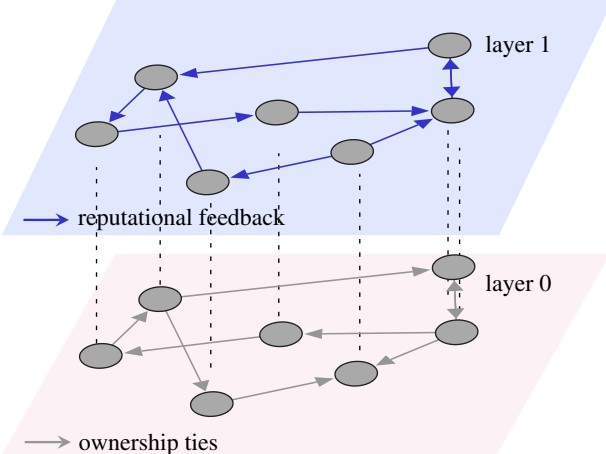

**Figure 1.** Illustration of the multilayer network consisting of two layers that capture ownership relations (layer 0) and reputation relations (layer 1). Note the *directionality of the links*: reputation spillovers occur from a firm to its shareholding companies through links with a direction opposite to the direction of ownership. In the example, because of cross-shareholding, every firm is a *direct* or *indirect* owner of all other firms. Hence, reputation spillovers will impact *all* firms either directly or indirectly.

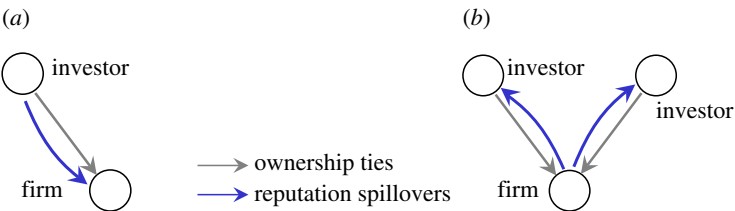

**Figure 2.** The direction of reputation spillovers changes with the formation of ownership structure. (*a*) In the first stage, reputation spillovers happen in the same direction as ownership relations. (*b*) In the second stage, reputation spillovers happen in the direction opposite to ownership relations.

firms, whereas *layer 1* contains their *reputation relations*. The reconstruction of *layer 0* will be discussed in detail in the Methods section. In the following section, we explain how reputation relations are related to ownership relations.

## 2.3. Reputation relations along ownership

We start from the insight that, in the multilayer network framework, the nodes in network layers 1 and 0 are identical. That means, the firms are the same, but their ownership relations, depicted in layer 0, differ from their reputation relations, depicted in layer 1. To assess these reputation relations, in line with the existing literature, we first assume that *reputation spillovers* between firms can be channelled through ownership relations. Based on the *signalling theory* [10] which explains how organizations communicate and interpret signals, we further argue that the *direction* of reputation spillovers changes during the formation of ownership structure, as explained in the following.

Concretely, as figure 2 illustrates, in the first stage of ownership formation, institutional shareholders start to hold shares of a company. Their shareholding decisions are based on careful screening and thus create a signal to other potential investors. These signals become stronger with the reputation of the early investors. That is, during the first stage early shareholders 'spread' their reputation to the invested company. Such signals reduce the information asymmetry between the earlier shareholders and other potential investors. Specifically, these signals can attract potential investors. If they interpret these signals in a positive manner, they may decide to invest in the same company, as well [25]. This leads to our

**Assumption 2.1.** *When institutional investors start to invest in a company, the direction of reputation spillover is from the investors to the invested companies.*

While during the first stage reputation spillover occurs in the direction of ownership, this effect does not last forever. After the invested company has obtained a rather stable ownership structure with a fixed number of institutional shareholders, holding shares of this company will become a status symbol that

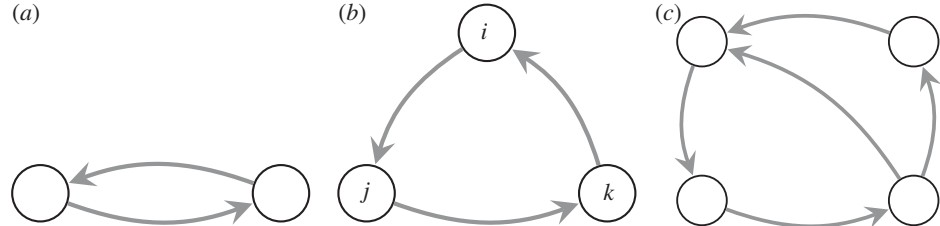

**Figure 3.** Cross-shareholding structures of (*a*) two firms, (*b*) three firms and (*c*) four firms.

signals reputation feedback to the shareholders. As a positive example, the increase in the reputation status of a firm comes with market signals on the firm's activities and prospects. One of these signals is the change in its dividend payout, which further increases the potential profit of its shareholding companies as well as their reputation. As a negative example, a firm's reputation damage can be attributed to the poor quality of its shareholders' monitoring performance, which in turn signals negative reputation spillovers to its shareholders.

**Assumption 2.2.** *When a company has a stable ownership structure, the direction of reputation spillover is mainly from the company to its shareholders.*

Although no studies have explicitly tested the different directions of reputation spillover in the formation of ownership structure, we can find indirect support for our two assumptions. Concretely, changes in the direction of reputation spillover can be seen as a special case of the Simmel effect [26], which originally describes how status symbols, i.e. externally displayed traits associated with high social class, gain their popularity among social agents. Once such symbols spread through the population, the Simmel effect states that these symbols have little value to the agents and thus are bound to lose their popularity.

In the case of ownership, shares could play the role of status symbols, their cost and value, however, does not decrease but increases over time. For the first shareholders of a firm, opportunity costs play the major role. As more institutional shareholders become signalled and invest in this firm, the increasing costs to align the interests of existing shareholders become more important. Eventually, when the cost of shareholding is high enough for newcomers, there will rarely be new shareholders and the company has a rather stable ownership structure.

So far, we have argued how reputation relations are related to ownership relations. In the following, we describe how we construct an empirical ownership network from data, and propose a dynamics for the reputation of firms in the stable core of this network.

# 3. Methods

## 3.1. Data

Information on the ownership structure of firms is available from different, mostly commercial, databases. We use the ORBIS database of 2007, which contains 37 million persons and firms with information on firms' location and operating revenue, and 13 million ownership relations with information on the percentage of ownership. Hence, the directly available information about firm $i$ is its location, operating revenue and ownership relations.

From the data about individual firms, we construct the global ownership network. This is a non-trivial task, as it also needs to consider *indirect* ownership, i.e. if firm $i$ holds a share of 10% on firm $j$ and $j$ holds 20% on firm $k$, then $i$ indirectly also owns a share on $k$ which maps to some influence of $i$ on $k$, respectively. This problem is exacerbated by the fact that $k$ can also hold a share on $i$ as figure 3*b* indicates, a phenomenon knows as cross-shareholding.

To overcome the above problem, we start by identifying a list of transnational companies. These are powerful players that shape and dominate the world economy, and the global ownership network should be centred around these companies. Merging companies which stay in a multinational group into one transnational company, we have compiled a list of 43 060 transnational companies located in 116 countries. The global ownership network of these transnational companies is constructed by recursively including all companies who are participated in by transnational companies, or companies

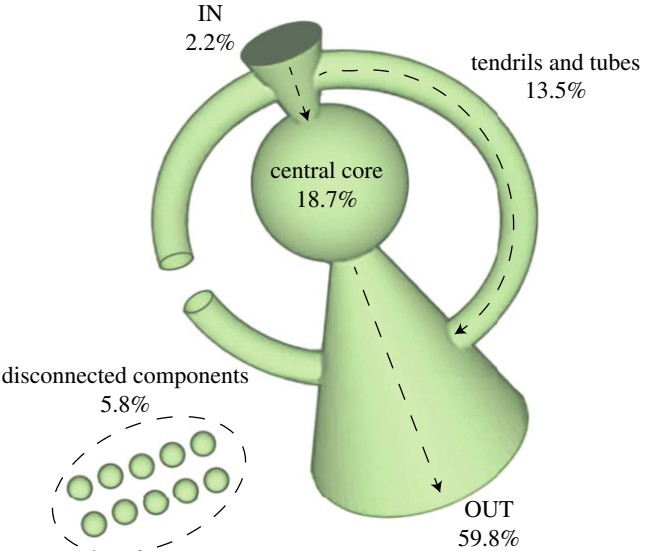

**Figure 4.** A schematic picture of the bow-tie structure of the global ownership network. The volume of each named group scales logarithmically with the share of operating revenue (numbers given). The dashed line with an arrowhead highlights the direction of ownership among the different groups.

who are shareholders of transnational companies, directly or indirectly. With this procedure, we end up with a network that contains 600 508 economic entities connected by 1 006 987 ownership relations, overall.

Previous works on the global ownership network [23,27] have revealed an interesting macroscopic pattern, known as a *bow-tie* structure which is shown in figure 4. The majority of companies belongs to this bow-tie, which is a *single connected component* in which four groups of firms can be identified. The first group is the *central core* in which companies own each other directly or indirectly via cross-shareholding relations [28] (see figure 3 for examples). Second, there is the IN *group*, which contains companies that are only shareholders of firms in the core. Group three is the OUT *group*, which, opposite to the IN group, contains companies of which firms in the core are the shareholders. And the last group, named *tendrils and tubes*, consists of companies that are separated from the core but are only shareholders of firms in the IN or the OUT group.

From the four groups, we focus only on the *central core* of the ownership network for the following two reasons: first, this core contains 1318 companies in 26 countries, interconnected by an overall number of 12 184 directed links. Despite its small size as compared to the whole network, this core can be seen as an important *super-entity* that owns 40% of all the transnational companies [23] in the world. Second, inside this core, every company is a direct or indirect shareholder of all others. Companies form cross-shareholding relations often because of their long-term strategies to avoid takeovers, reduce transaction cost, share information and increase trust. As a result, the ownership relations inside this core are more stable, which implies they are more costly to change, than other parts that are loosely integrated in the whole network. This is also confirmed by empirical analysis that cross-shareholding relationships tend to be stable over the years [29,30].

## 3.2. Reputation dynamics

To model the reputation dynamics in relation to ownership links, we adopt the multilayer network approach already shown in figure 1. As explained in §2.3, layer 0 contains a static and stable ownership network where firms are connected by directed ownership relations. Layer 1 captures reputation spillovers occurring from a firm to its shareholding companies through a directed link with the opposite direction of ownership (see assumption 2.2).

We denote the reputation of firm $i$ as $x_i$ and the reputation of its $k$ invested companies as $x_{i1}, x_{i2}, \ldots x_{ik}$. Then, the reputation dynamics for firm $i$ can be described with the general form: $\dot{x}_i(t) = f(x_{i1}, x_{i2}, \ldots x_{ik}, x_i, t)$. That is, $\dot{x}_i(t)$, the change in the reputation of firm $i$ at time $t$, does not only depend on its own reputation, but also on the reputation of firms that $i$ holds shares of. The functional form of $f$ should intuitively capture the following assumptions about reputation: First of all, reputation

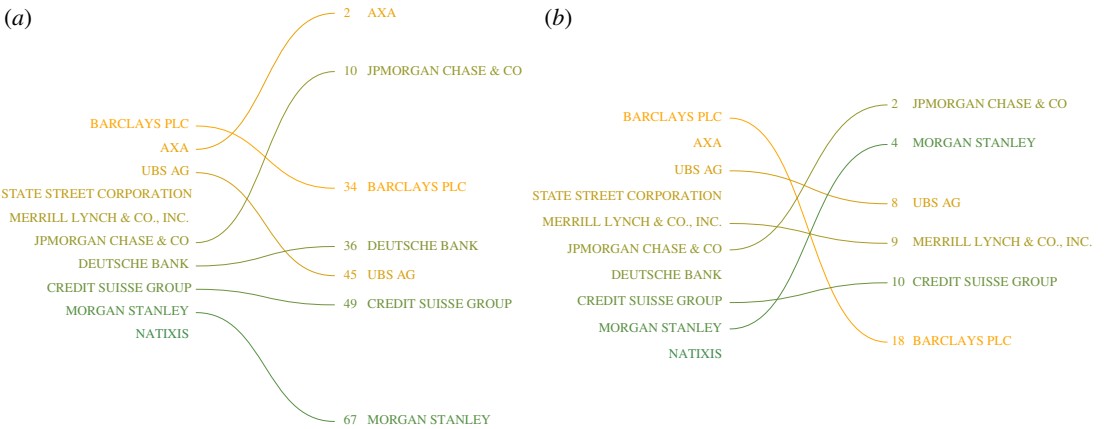

**Figure 5.** List of firms ranked by (left) ownership-based reputation and (right) by brand-based reputation (*a*) and employee-based reputation (*b*). Colours from dark green to light orange encode increasing values of ownership-based reputation of firms.

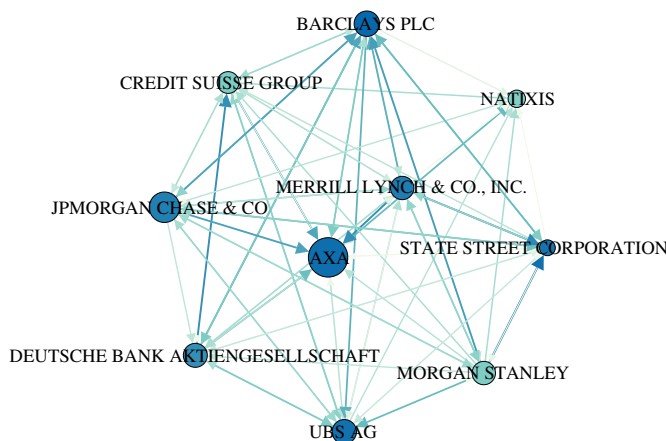

**Figure 6.** Ownership network of the top 10 reputable firms. The node size is proportional to the logarithm of the operating revenue of the firm, the node colour codes their calculated reputation values at equilibrium (from light to deep blue).

has to be maintained, that means, a continuous effort is needed to just keep the current level of reputation, if not to increase it. If a firm no longer invests in other firms, its reputation from ownership will decrease over time. Second, if a firm keeps its shares in other firms, it will receive a compensation in terms of reputation increase. This compensation should be larger from invested firms with a higher reputation.

As a proof of concept, a linearized form of *f* is described in equation (A 1) of the appendix. This dynamics takes into account an idea that is similar to PageRank, an algorithm to estimate the importance of a website by aggregating the importance of websites pointing to it [31]. Given enough time, starting from a random initial state, this dynamics converges to a non-trivial equilibrium. In the following, we use this equilibrium value to proxy the ownership-based reputation of firms.

## 4. Results

Here, we seek to demonstrate that the ownership relations of firms indeed have a reasonable validity for quantifying their reputation. To achieve this, we first rank firms using their ownership-based reputation calculated as described in the appendix, to then concentrate on the top 10 most reputable firms. These are listed on the *left-hand side* in figure 5, with increasing values of reputation colour-coded from dark green to light orange. We observe that the 10 most reputable firms are *financial institutions* for which, indeed, their reputation accounts for a large part of the total asset.

Figure 6 illustrates the ownership network among these ten firms. Each *directed link A → B* represents an ownership relation, from owner *A* to owned firm *B*. We note that the network is rather dense, which is

in line with the discussion about the central core in §3.1. This implies that all ten firms own each other, either directly or indirectly. Moreover, it can be seen that mutual ownership, $A \leftrightarrow B$, is quite common. The *size* of the nodes encodes the operating revenue of the firms, whereas the *colour* of the nodes encodes their reputation value. We emphasize that the firms with the highest reputation are also the ones with the largest number of incoming links, i.e. the largest number of shareholders.

We compare our results with reputation rankings perceived by different stakeholders. Ideally, our ranking should be compared with rankings that cover *all firms* in the core of the ownership network. However, to the best of our knowledge, such rankings are not available, because most rankings on reputation only provide information on a small number, between 50 and 100, of top-ranked firms or brands at the country level or industry level. Often it is unclear how many firms with statistically valid data are ranked in total; therefore, we cannot statistically compare our ranking against other rankings.

To demonstrate the usability of our approach, we only focus on the overlap of the most reputable firms in our ranking and two different alternative rankings: (i) the list of *Top 100 Financial Services Brands in the World*, i.e. the reputation perceived by *customers*, as provided by a brand management consultancy,[1] and (ii) the list of *eFinancialCareers Ideal Employer Rankings 2018*.[2] This list tracks the financial institutions that professionals most want to work for and admire, and focuses on reputation perceived by *employees*. The results are shown in figure 5. For both empirical rankings, on the left-hand side, we list the top 10 reputable firms according to our ownership-based reputation value. On the right-hand side, we list the position of these 10 firms in the *customer-based reputation* and in the *employee-based reputation*.

First of all, we note a remarkable overlap between our ownership-based reputation ranking and the reputation ranking obtained from two very different sources of information: (i) firm reputation as perceived by *customers* and (ii) firm reputation as perceived by (potential) *employees*. In the first case 7 and in the second case 6, out of our 10 firms appear in the alternative rankings. Why is this remarkable? Recall that our ranking was produced based on a network approach that only considers (a) the ownership structure of firms (*topology*) and (b) a feedback between the reputation of a firm and its owners (*dynamics*). We have not evaluated specific information about the firms, such as their performance, market capitalization or financial stability. Secondly, we have determined our ranking based on *1318 firms*, thus it is far from trivial that the top 10 firms of our reputation ranking are also among the top firms in other rankings. This demonstrates that using the interdependence between reputation and ownership, we can have a proxy of corporate reputation that identifies the most reputable firms.

One could still argue that our measure of ownership-based reputation does not provide additional insights, as it may in fact correlate with other information about firms, for instance, their operating revenue or their control power on other firms. That would mean, if firms already appear as important according to some economic indicators, they also rank high in our reputation ranking, and we cannot learn anything new from this. To investigate this conjecture, we have statistically tested the correlation between the ownership-based reputation of firms and (a) their operating revenue and (b) their integrated control, for the whole set of all 1318 firms. Integrated control is an economic measure of the influence exerted by a firm based on ownership relations. It is calculated by aggregating the shares held directly and indirectly by a firm [23]. The results are shown in figure 7.

Indeed, we observe a positive correlation between our reputation measure and the operating revenue of firms. This can be attributed to the 'halo effect' of financial performance on reputation [32,33]. This effect has been shown to explain about 50% of the variance of the overall rating of reputation measured in the Fortune survey [11]. The same positive correlation can be also found for our reputation measure and the integrated control of firms. This, however, is to be expected because both ownership-based reputation and integrated control are calculated from the same dataset and use the same information about ownership.

A second look at the scatter plots shown in figure 7, however, reveals that the distributions are *very broad* and the Kendall $\tau$ correlations are *very low*. Thus, we are not allowed to overinterpret these positive correlations. In fact, we observe a non-negligible amount of *outliers* in the top-left corners of both diagrams, i.e. firms with high operating revenue or high integrated control can still have a low reputation, according to our measure. On the other hand, we also observe that firms with a smaller operating revenue can have high reputation values. Hence, we conclude that our ownership-based reputation measure provides *additional information* that cannot be fully captured by economic measures such as integrated control or operating revenue.

---

[1]http://bankinnovation.net/2009/04/the-top-100-financial-services-brands-in-the-world/

[2]https://www.efinancialcareers.com/ideal-employer/location/global

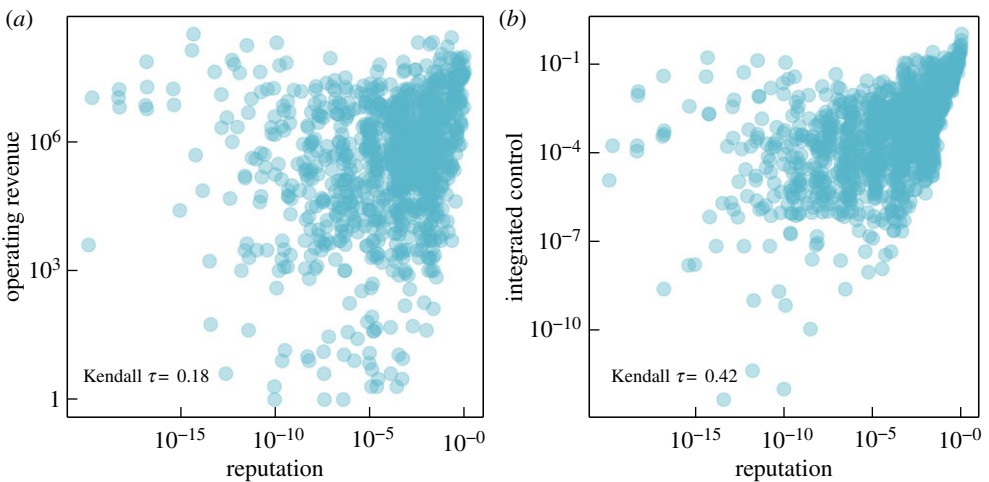

**Figure 7.** Scatter plot to relate the reputation of firms to their operation revenue (*a*) and their integrated control (*b*). The Kendall $\tau$ coefficients for these two comparisons are quite low (0.18 and 0.42), which indicates that the reputation of firms cannot be simply deduced from these two economic measures.

# 5. Discussion

How can reputation as an intangible asset be quantified in a more objective and more algorithmic manner? In our paper, we propose a novel way to reach this goal. Instead of using surveys or proxies for reputation, such as performance measures, we leverage available information about the *ownership structure* of firms. With reference to the signalling theory, we provide arguments of how ownership and reputation are linked from a theoretical perspective.

While there is support for our assumption that ownership links also channel reputation spillovers, we further elaborate on the *direction* of these spillovers. We distinguish between two phases related to the formation of the ownership network: (i) in the early phase, the ownership structure is still evolving and reputation is channelled from institutional investors to the invested company, (ii) in the late phase, the ownership structure is consolidated and reputation is channelled from the invested company back to the investor, that means in a direction opposite to the ownership link. Our quantification of reputation focuses on this late stage, i.e. we assume a stable ownership network. This does not imply that ownership relations never change; instead, it means that they change at a time scale that is much longer than the time scale of the reputation spillovers. Precisely, with respect to these spillovers we can assume a static ownership network.

To reconstruct this network requires a considerable effort because firms, in addition to direct investments, also indirectly participate in other firms via intermediaries. For this cumbersome calculation, we restrict ourselves to the ownership structure of 43 060 transnational companies and their participated companies, i.e. in total 600 508 firms. The resulting ownership network with 1 006 987 *directed and weighted links* consists of a small, but densely connected central core of 1318 firms which own each other either directly or indirectly via 12 184 ownership links. To illustrate how our measure of ownership-based reputation works, we concentrate on this small core.

The main idea behind our quantification of reputation is to model the *reputation spillover* between a firm and its investors. We propose a reputation dynamics which assumes that this spillover is proportional to the reputation of the firm, weighted by the ownership share. Recent incidents about reputation losses in the industry support this assumption. In Germany, the emission scandal of the car building company Volkswagen led to a *negative reputation spillover* to its investors, among them other car building companies such as Porsche, but also institutional investors. This drop-down of the investors' reputation was the larger, the higher the initial reputation of the invested company was. Additionally, in our dynamics, we assumed that reputation decays over time and therefore has to be constantly maintained.

This reputation dynamics has to be solved in a self-consistent manner, i.e. as a system of $N$ coupled *first-order differential equations*, where $N$ is the number of firms considered. It converges to an equilibrium much faster than changes of the ownership network. Thus, we can use these equilibrium values to quantify the ownership-based reputation of firms.

We evaluated the validity and usability of our reputation measure in two ways. First, we compared the reputation ranking of firms based on our measure with alternative rankings that use customer or employee surveys. We found that, out of 1318 firms considered, the alternative rankings listed 6–7 out of the 10 top-ranked firms, we identified. This means that our ownership-based reputation measure is able to correctly detect firms with high reputation, that have been identified so far only by selective case studies of limited size. Secondly, we verified that our reputation measure is not a simple transformation of economic information about firms, such as operating revenue or integrated control. While correlations to these two variables exist, their Kendall $\tau$ coefficient is quite low. This means, our ownership-based reputation measures provides additional insights.

To further characterize our approach of quantifying reputation, we highlight that it can be quite easily implemented and automatically processed. This marks a big advantage in comparison to traditional approaches, which extensively rely on survey data to proxy corporate reputation. Our approach can be further extended if the ownership structure changes, as long as the assumption holds that reputations spillovers occur faster than changes of ownership relations. This is valid in particular for the core of the ownership network, which contains the most important firms with rather stable ownership relations. To fully model the *coevolution* of ownership relations and reputation spillovers would obviously require a more elaborated framework, which can still be based on the two-layer network model we proposed here. However, this framework would additionally require information on the current or initial value of reputation, as well as calibration of the time scale in which reputation changes with ownership. With the novel way of quantifying reputation based on ownership, our network approach is a promising step forward to overcome the limitations of survey-based approaches and a complementary way to quantify reputation.

Data accessibility. The datasets supporting this article have been uploaded as part of the supplementary material.
Authors' contributions. Y.Z. and F.S. designed and performed the research. Y.Z. analysed the data and computed the results. F.S. and Y.Z. wrote the text and gave final approval for publication.
Competing interests. We declare we have no competing interests.
Funding. We received no specific funding for this work.

# Appendix A

Reputation dynamics depends on many factors, such as the business type of firms or constraints imposed by legal regulations. The focus of this paper is to build on the interdependence between ownership and reputation, as shown in figure 1. Therefore, we treat the reputation of firm $i$ as a scalar variable $x_i(t)$ which can change on a short time scale, to capture reputation spillovers from other firms $1, \ldots, k$. The general dynamics $\dot{x}_i(t) = f(x_{i1}, x_{i2}, \ldots x_{ik}, x_i, t)$ requires us to determine a functional form for $f$. We propose the following linear dynamics, which can be considered as a first-order approximation of $f$ for a limited time window:

$$\dot{x}_i(t) = \sum_{j=1}^{N} a_{ji}(t)x_j(t) - \phi x_i(t), \tag{A 1}$$

The coefficients $a_{ji}$ represent the strength of reputation spillovers from $j$ to $i$ at time $t$. Our main modelling assumption is that reputation spillovers are channelled through ownership links. Therefore, $a_{ji} \neq 0$ if $i$ is a shareholder of $j$, i.e. $j \in \{i_1, i_2, \ldots, i_k\}$, if there are $k$ shareholders of $i$.

The linear equation (A 1) captures two key assumptions: First, a firm's reputation depends on the reputation of firms it invests in. Particularly, we assume for the strength of reputation spillovers the following log-linear form: $a_{ji} = \log(c \cdot w_{ij})$, in which $c$ is a normalization factor such that the minimum value of $a_{ij}$ equals one. The $w_{ij}$ denote the portion of shares firm $i$ holds in $j$ and can be obtained empirically from ownership databases such as ORBIS [27]. We note that we have also tested other than the log-linear form for $a_{ji}$, combining factors such as the operating revenue of firms, the number of ownership relations a firm has. These modifications, however, do not change our main results.

The second key assumption captured in equation (A 1) states that, without holding shares in other companies, the ownership-based reputation of one isolated firm cannot be maintained and decays with a constant rate of $\phi$. Obviously, the larger $\phi$ is, the faster the reputation decays, and the more effort it takes to keep the reputation above a certain level. Here, we simply assume that the decaying rate in reputation is the same constant for every firm in the network.

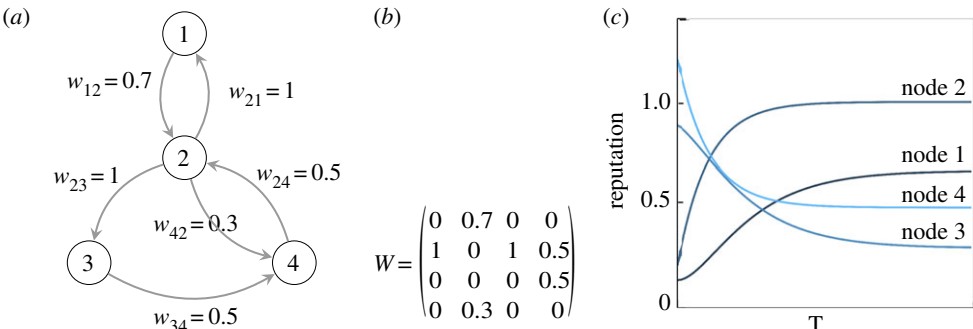

**Figure 8.** (a) Ownership network of four firms and (b) the corresponding weighted matrix **W** reflecting their ownership shares. (c) Numerical calculation of the reputation dynamics, to confirm the equilibrium values obtained analytically.

Writing equation (A 1) for each firm $i$, we have a linear dynamical system of coupled first-order equations. In matrix form, this set of equations becomes

$$\dot{\mathbf{X}}(t) = \mathbf{A}^{\mathrm{T}}X(t) - \phi\mathbf{X}(t), \tag{A 2}$$

with the vector $\mathbf{X}(t) = [x_1(t), x_2(t), \ldots, x_N(t)]^{\mathrm{T}}$ (where the transpose indicates that it should be a column vector rather than a row vector) and $a_{ji}$ as the elements of the matrix **A**. $\mathbf{A}^{\mathrm{T}}$ is the transposed matrix of **A**, $[\mathbf{A}^{\mathrm{T}}]_{ij} = [\mathbf{A}]_{ji}$, i.e. the matrix **A** is reflected over its diagonal such that the elements $a_{ji}$ have switched their row and column positions. Given an initial condition $\mathbf{X}(0) = [x_1(0), x_2(0), \ldots, x_N(0)]^{\mathrm{T}}$, the dynamics of $\mathbf{X}(t)$ is a function of $\mathbf{A}^{\mathrm{T}}$. An equilibrium exists only if $\mathbf{A}^{\mathrm{T}}X(t) = \phi\mathbf{X}(t)$, in which $\phi$ is an eigenvalue of $\mathbf{A}^{\mathrm{T}}$. Since $\mathbf{A}^{\mathrm{T}}$ is a non-negative matrix, the *Perron–Frobenius theorem* tells us that the eigenvector $\mathbf{X}^{\mathrm{eq}} = [x_1^{\mathrm{eq}}, x_2^{\mathrm{eq}}, \ldots, x_N^{\mathrm{eq}}]^{\mathrm{T}}$ corresponding to the largest eigenvalue of $\mathbf{A}^{\mathrm{T}}$ contains only positive entries. The entries $x_i^{\mathrm{eq}}$ depend on the initial conditions $\mathbf{X}(0)$ In order to allow a comparison between different set-ups, we rescale the eigenvector $\mathbf{X}^{\mathrm{eq}}$ using $r_i = x_i^{\mathrm{eq}}/x_{\mathrm{max}}^{\mathrm{eq}}$, where $x_{\mathrm{max}}^{\mathrm{eq}}$ is the largest entry. This leads us to the vector of relative reputation values $\mathbf{R}^{\mathrm{eq}} = [r_1^{\mathrm{eq}}, r_2^{\mathrm{eq}}, \ldots, r_N^{\mathrm{eq}}]^{\mathrm{T}}$. This information is used in the paper to rank firms with respect to their ownership-based reputation.

As a small didactic example, figure 8a sketches an ownership network of *four* firms. The corresponding matrix **W** with the weighted ownership links $w_{ij}$ is shown in figure 8b. The respective values for $a_{ji} = \log (c \cdot w_{ij})$ are obtained from the log-linear transformation (note the change of direction). We can then apply the equilibrium condition, $\mathbf{A}^{\mathrm{T}}X(t) = \phi X(t)$, to obtain the largest eigenvalue of 2.05. The corresponding eigenvector is $\mathbf{X}^{\mathrm{eq}} = [0.50, 0.75, 0.22, 0.34]^{\mathrm{T}}$. Rescaling this eigenvector with $x_{\mathrm{max}}^{\mathrm{eq}} = 0.75$ gives the vector of relative reputation values $\mathbf{R}^{\mathrm{eq}} = [0.67, 1, 0.29, 0.49]^{\mathrm{T}}$.

To confirm that the reputation dynamics, equation (A 1), indeed converges to the equilibrium values given, we run numerical calculations shown in figure 8c. We fixed $\phi = 2.05$ which is the largest eigenvalue of the weighted matrix $\mathbf{A}^{\mathrm{T}}$. For the initial condition $\mathbf{X}(0)$ we assigned random positive values to the four firms. Equation (A 1) is then calculated using the Runge–Kutta method. As shown in figure 8c, the equilibrium values are reached very fast and are consistent with entries in the vector $\mathbf{R}^{\mathrm{eq}}$. We note again that, if we set the damping factor $\phi$ to be the largest eigenvalue of the weighted matrix $A^{\mathrm{T}}$, we can use the analytic solution instead of numerical calculations.

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
