## [Reviewer comments · Royal Society Open Science]

Review History

RSOS-190570.R0 (Original submission)

Review form: Reviewer 1 (Markus Kirkilionis)

Is the manuscript scientifically sound in its present form?

Yes

Are the interpretations and conclusions justified by the results?

Yes

Is the language acceptable?

Yes

Is it clear how to access all supporting data?

Yes

Do you have any ethical concerns with this paper?

No

Have you any concerns about statistical analyses in this paper?

No

Recommendation?

Accept as is

Comments to the Author(s)

None, see attached file (Appendix A).

Review form: Reviewer 2

Is the manuscript scientifically sound in its present form?

No

Are the interpretations and conclusions justified by the results?

No

Is the language acceptable?

No

Is it clear how to access all supporting data?

Yes

Do you have any ethical concerns with this paper?

No

Have you any concerns about statistical analyses in this paper?

No

Recommendation?

Major revision is needed (please make suggestions in comments)

Comments to the Author(s)

The authors propose a way to measure the reputation of firms by using direct and indirect ownership links. They show that their measure correlates with other reputation measures for a list of top companies.

I have major concerns on the contribution:

1) The authors do not define what reputation is for them. They do provide a literature review where corporate reputation has been studied one way or another. They do compare their findings with a notion of brand-reputation and employee-based reputation. Yet, it is not clear to me what is the new notion of reputation that they are comparing from their findings. I can only understand that it should be 'something' that can be passed along ownership networks but in the opposite direction.

2) The assumptions are not well grounded. Institutional investors can continuously invest and disinvest their funds in the equity of companies, based on their changing profitability. Hence, what is exactly a stable ownership structure? I cannot really find anything realistic in the idea that there is a moment when one starts investing and a moment when the ownership structure of

a company becomes stable. Companies that are quoted at the stock exchanges have a floating amount of stake, changing hands every day. This is the case when 'reputation' could matter more for daily investors in the share capital.

3) Finally, one could not take as a validation of the exercise the fact that (a portion of) some first top 10 firms can be found among other reputation indicators. Once the idea of what a reputation is clear, one may want to run a full correlation on the entire sample of about 1,300 companies.

4) The assumptions by the authors are dynamic in nature, with a "before" and an "after" on which the exercise is made. Yet the data are just a cross-section of year 2007. Why would you not use the time dimension of ownership changes?

I hope it helped!

Best regards

Decision letter (RSOS-190570.R0)

25-Jun-2019

Dear Professor Schweitzer,

The editors assigned to your paper ("The interdependence of corporate reputation and ownership: A network approach to quantify reputation") have now received comments from reviewers. We would like you to revise your paper in accordance with the referee and Associate Editor suggestions which can be found below (not including confidential reports to the Editor). Please note this decision does not guarantee eventual acceptance.

Please submit a copy of your revised paper before 18-Jul-2019. Please note that the revision deadline will expire at 00.00am on this date. If we do not hear from you within this time then it will be assumed that the paper has been withdrawn. In exceptional circumstances, extensions may be possible if agreed with the Editorial Office in advance. We do not allow multiple rounds of revision so we urge you to make every effort to fully address all of the comments at this stage. If deemed necessary by the Editors, your manuscript will be sent back to one or more of the original reviewers for assessment. If the original reviewers are not available, we may invite new reviewers.

- Data accessibility

If you wish to submit your supporting data or code to Dryad (<http://datadryad.org/>), or modify your current submission to dryad, please use the following link:
<http://datadryad.org/submit?journalID=RSOS&manu=RSOS-190570>

- Competing interests

- Authors' contributions

- Acknowledgements

- Funding statement

on behalf of Dr Robert MacKay (Associate Editor) and Mark Chaplain (Subject Editor)
openscience@royalsociety.org

Associate Editor's comments (Dr Robert MacKay):

Both reviewers considered this to be an interesting paper but one poses major questions that need to be addressed before it could be considered for publication. A major revision is invited.

Comments to Author:

Reviewers' Comments to Author:
Reviewer: 1

Comments to the Author(s)
None, see attached file

Reviewer: 2

Comments to the Author(s)
The authors propose a way to measure the reputation of firms by using direct and indirect ownership links. They show that their measure correlates with other reputation measures for a list of top companies.

I have major concerns on the contribution:

1) The authors do not define what reputation is for them. They do provide a literature review where corporate reputation has been studied one way or another. They do compare their findings with a notion of brand-reputation and employee-based reputation. Yet, it is not clear to me what is the new notion of reputation that they are comparing from their findings. I can only understand that it should be 'something' that can be passed along ownership networks but in the opposite direction.

2) The assumptions are not well grounded. Institutional investors can continuously invest and disinvest their funds in the equity of companies, based on their changing profitability. Hence, what is exactly a stable ownership structure? I cannot really find anything realistic in the idea that there is a moment when one starts investing and a moment when the ownership structure of a company becomes stable. Companies that are quoted at the stock exchanges have a floating amount of stake, changing hands every day. This is the case when 'reputation' could matter more for daily investors in the share capital.

3) Finally, one could not take as a validation of the exercise the fact that (a portion of) some first top 10 firms can be found among other reputation indicators. Once the idea of what a reputation is clear, one may want to run a full correlation on the entire sample of about 1,300 companies.

4) The assumptions by the authors are dynamic in nature, with a "before" and an "after" on which the exercise is made. Yet the data are just a cross-section of year 2007. Why would you not use the time dimension of ownership changes?

I hope it helped!

Best regards

Author's Response to Decision Letter for (RSOS-190570.R0)

See Appendix B.

Decision letter (RSOS-190570.R1)

02-Oct-2019

Dear Professor Schweitzer,

I am pleased to inform you that your manuscript entitled "The interdependence of corporate reputation and ownership: A network approach to quantify reputation" is now accepted for publication in Royal Society Open Science.

Best regards,

Lianne Parkhouse
Royal Society Open Science
openscience@royalsociety.org

on behalf of Dr Robert MacKay (Associate Editor) and Mark Chaplain (Subject Editor)
openscience@royalsociety.org

Associate Editor Comments to Author (Dr Robert MacKay):

Many thanks for your revised version and letter indicating how you have taken the reviewers' comments into account. I'm pleased to recommend it for publication now.

Appendix A

Review Manuscript RSOS-190570

Title: The Interdependence Reputation and Ownership: a Network Approach to Quantify Reputation.

Authors: Yan Zhang, Frank Schweitzer

The manuscript covers an important aspect of economics, especially the important ranking question, in this case the ranking of companies according to their reputation.

The paper is well suited for the RSOS Journal, the issue is of general interest. The text is well structured, the English is flawless, and can be printed more or less in its current state.

I do therefore recommend the publication in the RSOS Journal.

Of course, from the mathematical perspective, I have some questions, but these are only relevant for future development of the theory. Of course, one needs to start somewhere.

The mathematics of this paper is very simple, a combination of a 2-layer multilayer network, and a linear ODE, with the state variables being reputations, and the coefficients being driven or determined by the ownership relationship. This model is clearly too simple.

Some of the things I noted while reading: the model, after the ranking being applied, clearly has no temporal dimension. The ODE is thought to be a linearization, valid for short time scales. This totally neglects the history of the investment sequence, which clearly was already driven by reputation, at least partially. The page ranking algorithm is too simple, as it is one dimensional (ownership), there are now several multi-dimensional rankings available. The authors have tested modifications and claim this would not affect results. However, these perturbations were entirely in their own framework, and therefore this robustness is to be expected.

Moreover, the multi-scale dimension, both on the investor side, and on the company side, is totally neglected. It is a difference if a company is owned by one main shareholder, and many small ones, or whether the shareholders are evenly distributed. The companies have different sizes, which determines where they seek reputation, etc.

Final conclusion: a good paper, asking the right questions. But it can only be the beginning of further discussion.

Appendix B

Statement on the Revision of ⟨Paper ID⟩ Based on the Referees' Report

Author1

Author2

Author3

July 16, 2019

This statement concerns our revision of the ⟨Paper ID⟩ paper, entitled “⟨*Paper Title*⟩”, based on the referees' report.

Comments by Reviewer #1

1. The manuscript covers an important aspect of economics, especially the important ranking question, in this case the ranking of companies according to their reputation. The paper is well suited for the RSOS Journal, the issue is of general interest. The text is well structured, the English is flawless, and can be printed more or less in its current state. I do therefore recommend the publication in the RSOS Journal.

We would like to thank the reviewer for this strong recommendation.

2. Of course, from the mathematical perspective, I have some questions, but these are only relevant for future development of the theory. Of course, one needs to start somewhere. The mathematics of this paper is very simple, a combination of a 2-layer multilayer network, and a linear ODE, with the state variables being reputations, and the coefficients being driven or determined by the ownership relationship. This model is clearly too simple.

Some of the things I noted while reading: the model, after the ranking being applied, clearly has no temporal dimension. The ODE is thought to be a linearization, valid for short time scales. This totally neglects the history of the investment sequence, which clearly was already driven by reputation, at least partially. The page ranking algorithm is too simple, as it is one dimensional (ownership), there are now several multi-dimensional rankings available. The authors have tested modifications and claim this would not affect results. However, these perturbations were entirely in their own framework, and therefore this robustness is to be expected.

We agree with the reviewer that this model is a simple model that neglects the history of the investment sequence, which can be partially driven by reputation. To justify this simplification valid, we limit the analysis to the cross-shareholding structure of the global ownership network, which is much more stable and changes much slower than other parts of the network that are only loosely integrated in the network.

Indeed, there are now several multi-dimensional rankings available for corporate reputation. However, to the best of our knowledge, most of them only cover a small number of firms at the country level or industry level. Therefore, as the main goal of our paper, we start from a simple model to demonstrate the potentiality that ownership information can be used to quantify reputation of firms.

3. Moreover, the multi-scale dimension, both on the investor side, and on the company side, is totally neglected. It is a difference if a company is owned by one main shareholder, and many small ones, or whether the shareholders are evenly distributed. The companies have different sizes, which determines where they seek reputation, etc.

Thanks for this insightful suggestion. As reported by Delgado-Garcia et al. (2010), there is a negative correlation between the concentration of ownership in the largest shareholder and corporate reputation for firms in Spain. Therefore, as one possible next step, it is interesting to explore further whether the same pattern can be observed for firms in the core of the ownership network.

4. Final conclusion: a good paper, asking the right questions. But it can only be the beginning of further discussion.

We thank the reviewer again for his conclusion.

Comments by Reviewer #2

The authors propose a way to measure the reputation of firms by using direct and indirect ownership links. They show that their measure correlates with other reputation measures for a list of top companies.

I have major concerns on the contribution:

We would like to thank the reviewer for his/her careful reading of our manuscript. In our revised version we carefully considered all concerns and addressed all comments raised by the reviewer:

1. The authors do not define what reputation is for them. They do provide a literature review where corporate reputation has been studied one way or another. They do compare their findings with a notion of brand-reputation and employee-based reputation. Yet, it is not clear to me what is the new notion of reputation that they are comparing from their findings. I can only understand that it should be 'something' that can be passed along ownership networks but in the opposite direction.

Thanks for pointing this out. As we have described in the literature review, reputation is a multi-dimensional construct that aggregates various dimensions of different stakeholders' perceptions. In our paper, instead of providing a formal definition of reputation that aggregates all dimensions, we only focus on the dimension of reputation that relates to corporate shareholders, a key stakeholder group of companies connected by ownership relations. This is not a new notion of reputation, but rather one dimension of reputation that has been overlooked in the literature.

In the revised version, to make this point clear, we emphasize that our notion of reputation in both the second to last paragraph of page 2 and the last paragraph of page 3.

2. The assumptions are not well grounded. Institutional investors can continuously invest and disinvest their funds in the equity of companies, based on their changing profitability. Hence, what is exactly a stable ownership structure? I cannot really find anything realistic in the idea that there is a moment when one starts investing and a moment when the ownership structure of a company becomes stable. Companies that are quoted at the stock exchanges have a floating amount of stake, changing hands every day. This is the case when 'reputation' could matter more for daily investors in the share capital.

Thanks for this comment. Indeed, institutional investors can continuously invest and disinvest their funds in the equity of companies, based on their changing profitability. However, this kind of change in investment is much less likely and more costly to happen when companies own each other directly or indirectly. This kind of ownership is called cross-shareholding. As pointed out by the literature, one of the main purposes of cross-shareholding structure is in many respects for relationship building but not for investing for profits (Kanaya & Woo, 2001; Kaplan, 1997), and cross-shareholding relationships tend to be stable over the years (McGuire & Dow, 2003).

In the revised version, we add the literature to support our assumption in the model that cross-shareholding is considered a stable structure, in the last paragraph of subsection 3(a).

3. Finally, one could not take as a validation of the exercise the fact that (a portion of) some first top 10 firms can be found among other reputation indicators. Once the idea of what a reputation is clear, one may want to run a full correlation on the entire sample of about 1,300 companies.

Exactly, in the ideal scenario, we could run a full correlation between the different notions of reputation, or rather different dimensions of reputation for all of the 1,300 firms. However, to the best of our knowledge, most of the available rankings only cover a small number of firms at the country level or industry level. Therefore, we can hardly provide a full correlation analysis. As an alternative, we look at other properties of firms that can be comparatively easier to retrieve, such as operating revenue or integrated control.

In our draft, we emphasize the limitation of existing measures for reputation in the second to last paragraph of page 10.

4. The assumptions by the authors are dynamic in nature, with a "before" and an "after" on which the exercise is made. Yet the data are just a cross-section of year 2007. Why would you not use the time dimension of ownership changes?

Thanks for pointing this out.

Due to very costly data accessibility, we only have one snapshot of the ownership network in 2007. Therefore, in our work, we limit ourselves to the core of the ownership network which contains stable ownership relations that change less frequently than other parts of the ownership network. Despite this simplicity, owing to the stable nature of cross-shareholding relations, we have already demonstrated the interdependency between reputation and ownership, which is the main purpose of the paper.

To model the reputation dynamics of all firms in the ownership network, it is necessary to use the time dimension of ownership changes, which is not possible for us, due to the limited data accessibility. However, here we emphasize that even if we have the full sequence, we need further information on the current or initial value of reputation and calibration of the time scale in which reputation changes with ownership. Such information is currently not available. This is the long-term goal of our research in this line, towards which our work takes the first step.

In the revised version, we elaborate on this point in the discussion section.